# Expression and Role of Toll-like Receptors in Facial Nerve Regeneration after Facial Nerve Injury

**DOI:** 10.3390/ijms241411245

**Published:** 2023-07-08

**Authors:** Jae-Min Lee, Seung Geun Yeo, Su Young Jung, Junyang Jung, Sung Soo Kim, Myung Chul Yoo, Hwa Sung Rim, Hye Kyu Min, Sang Hoon Kim, Dong Choon Park

**Affiliations:** 1Department of Otorhinolaryngology—Head and Neck Surgery, College of Medicine, Kyung Hee University Medical Center, Seoul 02447, Republic of Korea; sunjaesa@hanmail.net (J.-M.L.); yeo2park@gmail.com (S.G.Y.); marslover@naver.com (H.S.R.); gprb9870@naver.com (H.K.M.); hoon0700@naver.com (S.H.K.); 2Department of Otorhinolaryngology—Head and Neck Surgery, Myongji Hospital, Hanyang University College of Medicine, Goyang 04763, Republic of Korea; monkiwh35@hanmail.net; 3Department of Anatomy and Neurobiology, College of Medicine, Kyung Hee University, Seoul 02447, Republic of Korea; jjung@khu.ac.kr; 4Department of Biochemistry and Molecular Biology, College of Medicine, Kyung Hee University, Seoul 02447, Republic of Korea; sgskim@khu.ac.kr; 5Department of Physical Medicine & Rehabilitation, College of Medicine, Kyung Hee University Hospital, Seoul 05278, Republic of Korea; famousir@naver.com; 6Department of Obstetrics and Gynecology, St. Vincent’s Hospital, The Catholic University of Korea, Suwon 442723, Republic of Korea

**Keywords:** facial nerve, degeneration, regeneration, toll-like receptor

## Abstract

Facial nerve palsy directly impacts the quality of life, with patients with facial nerve palsy showing increased rates of depression and limitations in social activities. Although facial nerve palsy is not life-threatening, it can devastate the emotional and social lives of affected individuals. Hence, improving the prognosis of patients with this condition is of vital importance. The prognosis of patients with facial nerve palsy is determined by the cause of the disease, the degree of damage, and the treatment provided. The facial nerve can be easily damaged by middle ear and temporal bone surgery, trauma or infection, and tumors of the peripheral facial nerve or tumors surrounding the nerve secondary to systemic disease. In addition, idiopathic, acquired immunodeficiency syndrome and autoimmune diseases may damage the facial nerve. The treatment used for facial paralysis depends on the cause. Treatment of facial nerve amputation injury varies depending on the degree of facial nerve damage, comorbidities, and duration of injury. Recently, interest has increased in Toll-like receptors (TLRs) related to innate immune responses, as these receptors are known to be related to nerve regeneration. In addition to innate immune cells, both neurons and glia of the central nervous system (CNS) and peripheral nervous system (PNS) express TLRs. A comprehensive literature review was conducted to assess the expression and role of TLRs in peripheral nerve injury and subsequent regeneration. Studies conducted on rats and mice have demonstrated the expression of TLR1–13. Among these, TLR2–5 and TLR7 have received the most research attention in relation to facial nerve degeneration and regeneration. TLR10, TLR11, and TLR13 increase during compression injury of the facial nerve, whereas during cutting injury, TLR1–5, TLR8, and TLR10–13 increase, indicating that these TLRs are involved in the degeneration and regeneration of the facial nerve following each type of injury. Inadequate TLR expression or absence of TLR responses can hinder regeneration after facial nerve damage. Animal studies suggest that TLRs play an important role in facial nerve degeneration and regeneration.

## 1. Introduction

Facial palsy, a peripheral nerve disorder resulting from facial nerve injury, is not life-threatening, but failure to completely recover from injury can lead to psychological, emotional, and social challenges that significantly impact patients’ social activities and quality of life [1,2]. Thus, achieving complete recovery is of the utmost importance. Although numerous studies designed to find a definitive cure for facial paralysis resulting from facial nerve damage have been conducted, there have been no significant breakthroughs to date.

Because the facial nerve has a longer course than other cranial nerves and passes through a narrow canal within the temporal bone, it can be easily damaged by middle ear and temporal bone surgery, trauma or infection, and tumors of the peripheral facial nerve or tumors surrounding the nerve. Such damage can lead to paralysis or facial nerve palsy secondary to systemic disease. In addition, facial paralysis can be caused by various factors, such as malignant otitis externa, tuberculosis, Lyme disease, acquired immunodeficiency syndrome, childbirth injury, iatrogenic injury, radiation injury, meningioma, histiocytosis, rhabdomyosarcoma, Mobius syndrome, lower lip palsy, recurrent facial palsy, Melkersson–Rosenthal syndrome, Guillain–Barre syndrome, and autoimmune diseases.

The treatment approach used for facial paralysis depends on the cause. In the case of Bell’s palsy, the most common cause of facial paralysis, high-dose steroids are administered within 3 days of facial paralysis onset. In the case of complete facial paralysis, combination treatment with steroids and antiviral drugs is effective. Treatment of eyes with incomplete eyelid closure due to facial paralysis is very important. Artificial tears or eye ointment should be used in the early stages, and, if pain or itching occurs, consultation with an ophthalmologist is necessary to prevent corneal damage. Physical therapy is also commonly used during the recovery period, although its effectiveness is not yet certain [3,4].

Surgery is also used to treat facial paralysis. In the case of Bell’s palsy, the most common form of idiopathic facial nerve palsy, facial nerve decompression has been performed on the hypothesis that inflammatory changes in the nerve due to viral infection cause facial nerve compression in the facial canal; however, the efficacy of using surgery to treat Bell’s palsy remains controversial. In patients with H-B grade VI complete paralysis, facial nerve decompression can be performed in patients who do not respond to drug treatment, in patients in whom motor fibers show below normal results in continuous nerve conduction examinations, and in patients in whom electromyography (EMG) examination shows abnormal potentials indicative of muscle denervation such as fibrillation potentials and positive sharp waves, provided there is no other medical disease or life-threatening trauma. In patients with iatrogenic facial nerve palsy that developed after otitis media surgery, reoperation should be performed as soon as possible if the facial nerve could not be identified during surgery or if there was an expected cut site during surgery. In patients with facial paralysis arising from temporal bone fracture, facial nerve decompression surgery is performed when there is complete paralysis at the time of the initial injury, when there is progressive facial paralysis, or when more than 90% of motor fibers have degenerated within 6 days on a nerve conduction test [5,6].

When the facial nerve is damaged, including the endoneurium, the facial nerve exhibits aberrant regeneration, ephaptic transmission, and cellular hypersensitivity during the regeneration process, resulting in abnormal facial movements and muscle contractions [7]. It is, therefore, not an exaggeration to say that the fate of a facial nerve injury is determined the moment the nerve is damaged. If there is a large percentage of facial nerve fibers with neurotmesis, the degeneration ratio increases relatively rapidly within a few days on electrophysiological examination. Patients with such nerve damage, which is classified as House–Brackmann V or VI, have a high risk of incomplete recovery. Compared with acute facial nerve injury, chronic compression injury has a different pathophysiological process. If chronic pressure is applied to the facial nerve, degeneration and regeneration of Schwann cells occur at the same time, and the function of the facial nerve is not greatly compromised. If, however, the compression continues for a long time, or if the pressure is high, the proportion of degenerative Schwann cells increases, resulting in reduced function of the facial nerve and facial paralysis [5,8]. Treatment of facial nerve amputation injuries varies depending on the degree of facial nerve damage, comorbidities, and duration of injury. In general, direct end-to-end anastomosis (i.e., directly connecting the nerve to the damaged site) has the best prognosis when the nerve is severed, and paralysis occurs within one month after nerve injury. If the length of the nerve defect is less than 1 cm, the anastomosis is performed by rerouting in the temporal bone without tension. If the defect length is greater than about 1 to 1.5 cm, a nerve graft is performed using a donor nerve such as the greater auricular nerve or sural nerve. If the time since the injury is 30 days to 1 year, or if the proximal facial nerve is unavailable, nerve substitution can be performed to help rehabilitate facial movements using other cerebral nerves [9]. In patients who have had facial nerve paralysis for more than 1 year, leading to severe facial muscle atrophy, muscle transposition is performed using the temporal muscle, masseter muscle, and digastric muscle [10,11]. 

When the cell body of a neuron is destroyed, the neuron can no longer survive. However, if an axon is partially severed, the neuron can regenerate the axon and, under appropriate conditions, the cell may re-synapse with other cells that were synapsing, allowing full function to be restored. In particular, peripheral nerve fibers can regenerate if there is no damage to the cell body. Changes in nerve fibers after nerve damage vary depending on the degree of damage. In the case of mild damage, such as neuropraxia, local demyelination and remyelination occur, whereas if severe damage occurs, degeneration and regeneration of axons occur [12,13]. Following nerve damage, a process known as Wallerian degeneration begins in the distal portion of the damaged nerve. Macrophages and Schwann cells, which dissolve and phagocytose the myelin sheet at the distal part of the damaged nerve, play an important role in this process. Among the byproducts generated by axon degeneration after nerve damage are endogenous damage-associated molecular patterns (DAMPs), which promote the expression of Toll-like signaling pathway, TLR2, -3, and -4, as well as MyD88 (myeloid differentiation primary response 88) in Schwann cells. These changes in expression are accompanied by the activation of secretion of tumor necrosis factor-α (TNF-α), interleukin-1 (IL-1), and monocyte chemoattractant protein-1 (MCP-1). MCP-1 stimulates phospholipase A2 (PLA2), which, in turn, stimulates lysophosphatidylcholine (LPC), directly promoting the degradation of myelin in Schwann cells. LPC also induces the expression of macrophage inflammatory protein-1α (MIP-1α) in peripheral nerve Schwann cells, which secrete IL-1β within 3 days of injury in vitro. IL-1β, MCP-1, and MIP-1α secreted from peripheral nerve Schwann cells on days 3–7 following nerve damage subsequently activate macrophages, which remove myelin by phagocytosis in vitro [14].

To understand the role of TLRs in nerve regeneration and associated changes in their expression and gain insight into the clearance of damaged nerve debris and innate immune responses to facial nerve damage, we conducted a review of the relevant literature. To this end, we analyzed and summarized the results of previous studies on the involvement of TLR in nerve regeneration. To date, studies on TLRs in nerves have mainly been conducted on components of the central nervous system, such as the spinal cord and brain, not on peripheral nerves, such as the sciatic nerve and facial nerve. Indeed, only a few studies have been conducted on the facial nerve. Therefore, through a literature review on peripheral nerves, the expression and role of TLRs in facial nerves are analyzed and summarized.

## 2. Toll-like Receptors

The most reliable way for a host organism to protect itself from pathogens is to accurately recognize their presence and initiate appropriate immune responses capable of incapacitating them. Pathogens are recognized by host organism receptors that recognize non-self-molecules present only in pathogens called pathogen-associated molecular patterns (PAMPs). Host organism’s immune responses are initiated and regulated not only by pathogens but also by microorganisms that form symbiotic/commensal relationships, a concept encapsulated in the broader, frequently used term, microbe-associated molecular patterns (MAMPs). Host immune responses are also induced by the recognition of self-materials generated by damage to host cells, substances called damage-associated molecular patterns (DAMPs). MAMPs and DAMPs are recognized by intracellular receptors in the host that are collectively referred to as pattern-recognition receptors (PRRs). The representative PRRs in animal cells are the well-known TLRs [15,16].

In the early 1980s, researchers discovered the gene encoding Toll in Drosophila, identifying it as an essential component of a signaling pathway that regulates dorsoventral polarity in early Drosophila embryos. This Toll receptor has subsequently been shown to be involved in the antifungal immune response of Drosophila [17]. Microbial infection in Drosophila stimulates the rapid production of various types of peptides with antimicrobial activity. The promoter regions of genes encoding these peptides, which are related to inflammatory and immune responses in mammals, contain nuclear factor-κB (NF-κB) binding sites. Drosophila Toll is involved not only in embryonic development but also in the immune response of adults [18,19]. Drosophila mutants in which Toll function is lost become highly susceptible to infection.

It is now known that TLRs are an evolutionarily conserved family of PRRs that play key roles in immune surveillance, priming antigen-specific adaptive immunity, and triggering innate immune responses to infectious pathogens in mammals. They also play important roles in inflammation, immune cell regulation, survival, and proliferation. Some TLRs are capable of recognizing certain endogenous molecules released by damaged cells or tissues after injury or under stress that serve as danger signals [20,21]. TLRs detect bacteria, viruses, fungi, and protozoa, triggering an innate immune response. Each TLR recognizes and is activated by a unique set of PAMPs, which act as ligands. TLR1, -2, and -6 ligands are mainly lipids, which are major components of bacterial cell walls and viral envelopes; TLR2, in particular, is activated by cell wall products of Gram-positive bacteria. In addition, TLR1 and -6 bind to TLR2 to mediate responses to microbial products. TLR3, -7, -8, and -9 ligands include bacterial and viral DNA (CpG DNA) and nucleic acids, such as RNA; activation of TLR3 by double-stranded RNA (dsRNA) leads to the activation of NF-κB [22]. TLR4 specifically recognizes and is activated by lipopolysaccharide (LPS), a cell wall component of Gram-negative bacteria [23], whereas TLR5 is activated by flagellin. TLRs are expressed in different cellular compartments, a difference that impacts which PAMPs are recognized [21]. TLR1, -2, -4, -5, and -6 are mostly expressed on the cell surface, whereas TLR3, -7, -8, and -9 are mainly expressed in intracellular compartments, such as endosomes [24]. 

TLRs play a crucial role in recognizing pathogens, and their deficiency can lead to various infectious, immune diseases owing to the failure to detect and respond to these pathogens. Moreover, abnormal TLR responses are associated with diseases such as otitis media, asthma, arteriosclerosis, autoimmune diseases, immunodeficiency, and sepsis. Though mainly present in immune cells, such as B cells, T cells, and macrophages, TLRs are known to be expressed in epithelial cells and vascular endothelial cells, and their expression has recently been reported in microglia, astrocytes, and neurons of the central nervous system. Notably, TLRs in the nervous system respond not only to PAMPs but also to endogenous DAMPs generated following nerve injury and are also activated by amyloid beta in the setting of Alzheimer’s disease. Importantly, ligand-activated TLRs are involved in signal transduction processes related to nerve damage and regeneration processes [25,26].

## 3. The Role of TLRs in Peripheral Nerve Degeneration and Regeneration (Figure 1) (Table 1)

In addition to innate immune cells, both neurons and glia of the CNS and PNS express TLRs. TLR-mediated neuroinflammation has been implicated in a variety of infectious and non-infectious neurological and neurodegenerative conditions of the CNS. TLR signaling modulates neural progenitor cell (NPC) proliferation and differentiation. Neurons produce a variety of cytokines and chemokines, partly through the direct activation of TLR signaling. The expression of functional TLR2–4 and TLR7–9 in neurons has been studied. Although research on TLRs in the peripheral nervous system is being conducted, studies on the facial nerve are scarce. The expression of TLRs 1–13 in the facial nerve after facial nerve injury has been reported; research on the roles of TLRs in facial nerve injury is ongoing.

**Figure 1 ijms-24-11245-f001:**
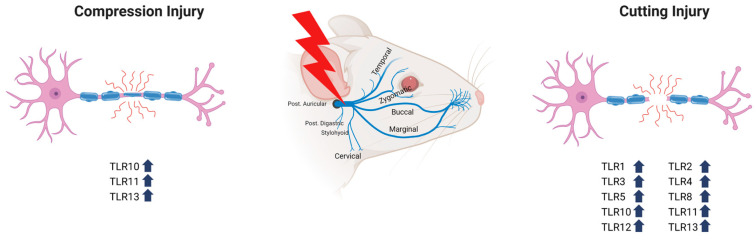
Increased expression of TLRs in the context of facial nerve degeneration and regeneration after facial nerve injury.

### 3.1. TLR2 and TLR4

TLR2 distinguishes itself from other TLRs by virtue of its recognition of diverse microbial components, such as PAMPs. Unlike other TLRs, which typically form homodimers, TLR2 sometimes forms a heterodimer with TLR1 or -6. It also functions in collaboration with Dectin-1, a receptor that recognizes fungal cell wall components. Notably, TLR2 can also recognize the form of LPS found in Gram-positive bacteria, such as streptococcus pneumoniae. The structure of this LPS is distinct from that of the LPS of Gram-negative bacteria recognized by TLR4. Studies have shown that Gram-positive bacteria cause septic shock in TLR2-deficient (Tlr2^−/−^) mice, establishing TLR2 as an important PRR in recognizing Gram-positive bacteria and initiating an immune response against them [27,28,29,30].

CD4^+^ Th2 lymphocytes are required for facial motoneuron (FMN) survival after facial nerve axotomy through interaction with peripheral antigen-presenting cells as well as CNS resident microglia. TLR2 is involved in the development of Th2-type immune responses and can be activated by intracellular components released from dead or dying cells. Immunization of mice with the TLR2 agonist Pam3Cys induces the Th2 cytokines, interleukin 5 (IL-5) and IL-13. Moreover, in a chronic asthma model, Th2 cytokine levels are significantly reduced in the lungs of Tlr2^−/−^ mice. Staphylococcus aureus, a bacterium that activates TLR2, increases the expression of the Th2 cytokines, IL-4, IL-5, and IL-13 in a mouse model of allergic conjunctivitis [31].

TLR4, the first TLR identified in humans, recognizes LPS, a cell wall component of Gram-negative bacteria, as well as endogenous ligands, such as HSP60, fibronectin, and hyaluronic acid, which are abundantly produced in response to stress. Although a small amount of LPS is capable of activatingTLR4, a large amount of endogenous ligands is required to activate this receptor [30]. Interestingly, when nerve damage occurs in both central and peripheral nerves, TLR4 inhibits the proliferation of neural progenitor cells (NPCs) and promotes neurodegeneration [32].

It has been reported that TLR2 and -4, known for their recognition of pathogens and regulation of immune responses, are expressed in adult NPCs and play key regulatory roles in neuronal differentiation in the hippocampus of adult rodents [33,34]. In particular, stimulation of TLR2 has been shown to accelerate the differentiation of NPCs into neurons without affecting their self-renewal capacity. In contrast, blocking the activation of NF-κB, a key effector of TLR2 signaling, significantly impairs the differentiation of astroglial cells. Conversely, stimulation of TLR4 blocks the differentiation of NPCs into neurons and attenuates their self-renewal capacity. These in vitro findings were further corroborated by in vivo experiments using mice genetically deficient for TLR2 or -4. These experiments showed that neural differentiation is impaired in mice with a genetic deficiency of TLR2, whereas both neural differentiation and reproduction are improved in mice genetically deficient for TLR4. Consequently, stimulation of TLR2 and -4 expressed in NPCs elicits both similar and different signal transduction processes. Both TLR2 and -4 signaling activates the intracellular protein, MyD88, leading to the phosphorylation of IκB kinase (IKK) and subsequent translocation of RelA into the cell nucleus. This process shares similarities in signal transmission in that both TLR2- and TLR4-induced signaling acts through a pathway involving the formation of an NF κBp50RelA complex. However, TLR4 can also engage an alternative, MyD88-independent signaling pathway that triggers phosphorylation of interferon regulatory factor 3 (IRF3). These differences in signal transduction suggest that TLR4 and -2 exert different regulatory actions during the differentiation of NPCs into neurons. An alternative interpretation is that different subsets of NPCs express either TLR2 or -4 and respond to specific stimuli accordingly. The brain shows a remarkable ability to protect itself from damage compared with other organs in the body. These findings thus suggest that the differentiation of NPCs into neurons or glia in response to TLR stimulation serves as a protective mechanism that minimizes damage to the central nervous system.

TLR2 and -4 expression has also been reported in the sciatic nerve, a type of peripheral nerve. In Tlr4^−/−^ (B6.B10ScN-Tlr4lps-del/JthJ) and Tlr2^−/−^ (B6.129-Tlr2tm1Kir/J) mice on a C57BL/6 background, a histomorphometric analysis following a standard sciatic nerve compression injury using Jeweler forceps revealed several adverse effects compared with C57Bl/mice. Specifically, both Tlr4^−/−^ and Tlr2^−/−^ mice showed fewer remyelinated nerves, more nerve debris, and diminished nerve regeneration after sciatic nerve compression injury compared with C57BL/6 mice [35], consistent with an important role for TLR2 and -4 in promoting nerve repair and recovery. In addition, Tlr4^−/−^ and Tlr2^−/−^ mice show reduced macrophage recruitment, persistent myelin debris in distal nerve stumps, and a significant delay in the process of Wallerian degeneration during nerve regeneration [36,37,38]. These results suggest that the impaired nerve regeneration observed in the absence of TLR2 or -4 is mainly attributable to a delayed demyelination process.

A preliminary microarray analysis conducted to assess changes in TLR2 and -4 mRNA levels after facial nerve injury revealed that TLR2 and -4 mRNA levels increased 142-fold and 4-fold, respectively, in the facial motor nucleus of WT mice on day 7 after axotomy [36]. A quantitative reverse transcription-polymerase chain reaction (RT-qPCR) analysis focusing on changes in TLR2 mRNA, which were much greater than those of TLR4, showed no significant difference in TLR2 mRNA expression between left uninjured (control) and right uninjured facial nuclei in WT mice. However, following facial nerve axotomy, TLR2 mRNA expression was rapidly and substantially increased in injured facial nuclei, reaching 194% ± 58%, 481% ± 36%, 893% ± 135%, and 1803% ± 38% of control values on days 1, 4, 7, and 14 post axotomy, respectively, before declining on day 30 (62% ± 38%; *p* < 0.01). Collectively, the results of this study suggest that TLR2 might be involved in the degeneration and regeneration of the facial nerve after facial nerve axotomy.

**Table 1 ijms-24-11245-t001:** Studies assessing TLR expression after facial nerve and sciatic nerve injury.

Author [Reference]	Study Design	Species and/or Sample	Nerve/Injury Methods	Detection Method	Target Gene(s) or Pathway(s) Associated with TLRs	Results/Conclusions
Wainwright DA, et al. [31]	Animal study	Seven-week-old female wild-type (C57BL/6) and *Tlr2*^−/−^ (C57BL/6 background) mice	Facial nerve/transection	qRT-PCR, immunofluorescence	TLR2	After facial nerve axotomy, TLR2 mRNA was significantly upregulated in the facial motor nucleus and TLR2 protein was co-localized to CD68^+^ microglia but not GFAP^+^ astrocytes. Studies using *Tlr2*^−/−^ mice revealed that TLR2 does not affect FMN survival after axotomy.
Lee H, et al. [32]	Animal study	*Tlr3*^−/−^ mice	Sciatic nerve/crush injury	qRT-PCR, flow cytometry, immunohistochemistry	TLR3	Nerve injury-induced axonal degeneration and subsequent axonal debris clearance were reduced in *Tlr3*^−/−^ mice compared with wild-type mice. Nerve injury-induced macrophage infiltration into injury sites was also attenuated in *Tlr3*^−/−^ mice, accompanied by reduced expression of the macrophage-recruiting chemokines, CCL2/MCP-1, CCL4/MIP-1β, and CCL5/RANTES. These data show that TLR3 signaling contributes to Wallerian degeneration after peripheral nerve injury by impacting Schwann cell activation and macrophage recruitment to injured nerves.
Wu SC, et al. [34]	Animal study	*Tlr2*^−/−^, *Tlr4*^−/−^ and C57BL/6 mice	Sciatic nerve crush injury	Western blot analysis, quantitative assessment of peripheral nerve architecture	TLR2, TLR4	A histomorphometric analysis revealed fewer remyelinated nerves and more nerve debris in both *Tlr4*^−/−^ and *Tlr2*^−/−^ mice than in C57BL/6 mice following sciatic nerve crush injury, indicative of worse nerve regeneration. Both TLR4 and -2 are crucial for nerve regeneration after nerve crush injury, mainly by delaying the demyelination, but not the remyelination, process.
Dubový P, et al. [39]	Animal study	Wistar rats	Sciatic nerve transection	Western blot analysis, qRT-PCR, immunohistochemical analysis	TLR9	Unilateral sciatic nerve lesions led to bilateral increases in levels of TLR9 mRNA and protein in both lumbar and remote cervical DRG compared with naive or sham-operated controls. These results suggest that a systemic innate immune reaction not only triggers a regenerative state in axotomized DRG neurons but also induces a pro-regenerative state further along the neural axis after unilateral nerve injury.
Hsieh CH, et al. [40]	Animal study	C57BL/6 mice; *Tlr2*^−/−^, *Tlr3*^−/−^, *Tlr4*^−/−^, *Tlr5*^−/−^, and *Tlr7*^−/−^ mice	Sciatic nerve crush injury	Semiquantitative immunohistochemical methods	TLR2, TLR3, TLR4, TLR5, TLR7	Tlr-knockout mice exhibited delayed expression of myelin genes and an altered expression pattern of myelination-related neurotrophin genes and transcription factors compared with C57BL/6 mice. Knockout of Tlr genes decreases the expression of myelination-related factors and impairs nerve regeneration after a sciatic nerve crush injury.
Min HK, et al. [41]	Animal study	Sprague–Dawley rats	Facial nerve/compression and transection	Vibrissae movement test qRT-PCR	TLR1, TLR2, TLR3, TLR4, TLR5, TLR6, TLR7, TLR8, TLR9, TLR10, TLR11, TLR12, TLR13	The scores for whisker movements in the cutting group were significantly lower than those in the crushing group. TLR9 and-13 mRNA expression levels were significantly lower in crush and cutting groups than in the control group on day 4 after injury. On day14 after injury, the expression of TLR2 mRNA was significantly higher in the cutting group than in the control group. TLRs may be involved in facial nerve damage and regeneration.
Min HK, et al. [42]	Animal study	Sprague–Dawley rats	Facial nerve/compression and transection	Vibrissae movement test/blinking reflex test, Western blotting	TLR1, TLR2, TLR3, TLR4, TLR5, TLR6, TLR7, TLR8, TLR9, TLR10, TLR11, TLR12, TLR13	Scores for whisker movements and blink reflexes in the crushing group showed improvement 14 days and 3 months after injury, whereas those in the cutting group were significantly lower at these time points. Western blot analyses showed that TLR11 and -13 were increased in the nerve-crush group, and TLR1, -2, -3, -4, -5, -8, -10, -11, -12, and -13 were increased in the cutting group after 14 days. After 3 months, TLR10 and -11 increased in the crushing group, and TLR1, -4, -5, -8, -11, and -12 increased in the cutting group. TLR1, -4, -5, -8, and -12 are involved in nerve degeneration after facial nerve injury, and TLR10, -11, and -13 are involved in recovery from facial palsy.

TLR, Toll-like receptor; qRT-PCR, quantitative RT-PCR; GFAP, glial fibrillary acidic protein; FMN, facial motoneuron; CCL, CC-chemokine ligand; MCP-1, monocyte chemoattractant protein-1; DRG, dorsal root ganglia.

### 3.2. TLR3

TLR3, which recognizes dsRNA produced during viral replication through a MyD88-independent pathway and triggers the production of type I interferon (IFN α/β) [18], also plays a role in other cellular processes. One such process is Wallerian degeneration, which is initiated following peripheral nerve injury and is known to involve Schwann cells. The involvement of TLR3 in Schwann cell activation during Wallerian degeneration was investigated using a loss of function approach. This study showed that mice lacking TLR3 (Tlr3^−/−^ mice) exhibit reduced axonal degeneration and clearance of axonal debris after sciatic nerve crush injury compared with wild-type mice. Moreover, nerve injury-induced macrophage infiltration into the injury site is attenuated in Tlr3^−/−^ mice, and the expression of macrophage-recruiting chemokines, CC-chemokine ligand (CCL) 2/MCP-1, CCL4/MIP-11β and CCL5/RANTES, is suppressed. Notably, these macrophage-recruiting chemokines were induced in primary Schwann cells in vitro by TLR3 stimulation. Collectively, these findings suggest that TLR3 signaling contributes to Wallerian degeneration by influencing Schwann cell activation and promoting macrophage recruitment to damaged nerves after peripheral nerve injury [43,44].

### 3.3. TLR9

In mammals, CpG motifs are usually heavily methylated; thus, the unmethylated CpG motifs of bacteria serve as PAMPs that can be recognized by TLRs. Specifically, following the engulfment of bacteria by macrophages and dendritic cells (DCs), unmethylated bacterial CpG DNA is degraded in the endosome, where it is recognized by endosomally-localized TLR9 [30].

One of the changes associated with Wallerian degeneration, a process distal to nerve injury, is the integration of axonal mitochondria and consequent leakage of mitochondrial DNA (mtDNA)—the natural ligand of TLR9. Following unilateral sciatic nerve compression or transection, RT-PCR, immunohistochemical, and Western blot analyses revealed changes in lumbar (L4-L5) and cervical (C7-C8) dorsal root ganglia (DRG). These analyses showed that TLR9 mRNA and protein levels were bilaterally increased in unilateral sciatic nerve lesions in the lumbar spine as well as in the distant cervical DRG compared with naive or sham-operated controls. This upregulation of TLR9 was associated with the activation of NF-κB and nuclear translocation of signal transducer and activator of transcription 3 (STAT3) in uninjured primary sensory neurons of the cervical DRG. This activation of the innate neuroimmune response is indicative of a pro-regenerative state. Thus, these results suggest that unilateral sciatic nerve lesions trigger a neuro-innate immune response through TLR9 and NF-κB, not only in the DRG associated with the injured nerve but also in the remote DRG [39].

### 3.4. TLR1–13

A number of recent studies on the role of TLRs in the facial nerve have explored multiple TLRs rather than focusing on one or two specific TLRs. In one study, nerve regeneration was tested in five knockout mouse lines (Tlr2^−/−^, Tlr3^−/−^, Tlr4^−/−^, Tlr5^−/−^, and Tlr7^−/−^) and C57BL/6 control mice (6 mice/group) subjected to a sciatic nerve crush injury, produced by continuous compression of the nerve for 30 s with Jeweler forceps [40]. Nerve regeneration following facial nerve injury was found to be compromised in all Tlr-knockout mice compared with C57BL/6 mice. In addition, expression of myelin genes was delayed and expression patterns of myelination-related neurotrophin genes and transcription factors were altered in Tlr-knockout mice compared with C57BL/6 mice. TLR signaling-mediated expression of insulin-like growth factor 2 (IGF2) and brain-derived neurotrophic factor (BDNF), as well as early growth response 2 (EGR2) and N-myc downstream-regulated gene 1 (NDRG1), were also significantly decreased in both early and late stages of nerve regeneration after crush injury in Tlr-knockout mice compared with C57BL/6 mice. Collectively, these observations indicate that knockout of TLR genes reduces the expression of myelination-related factors and impairs nerve regeneration after crush injury of the sciatic nerve [40].

Axotomy strongly induces peripheral nerve expression of TLR1, -2, -3, -4, and -7, which are involved in Schwann cell activation in response to endogenous danger signals. Stimulation of Schwann cells can also induce the expression of TLRs [37,45]. A previous study in rats investigated the expression of TLRs in Schwann cells, demonstrating that these cells highly express TLR2, -3, and -4 [46]. Another study also reported the expression of a wide range of TLRs (TLR1–9) in Schwann cells, with particularly high levels of TLR3, -4, and -7. In contrast, expression levels of these TLRs were remarkably low in motor and sensory neurons [38]. TLRs expressed in Schwann cells are functional and have been shown to increase the expression of pro-inflammatory mediators that contribute to the Wallerian degeneration process. The reduced presence of macrophages in Tlr2^−/−^ and Tlr4^−/−^ mice was shown to result in persistent myelin debris in the distal nerve stump and a significant delay in the Wallerian degeneration process during nerve regeneration [36].

The innate TLR response not only promotes clearance of inhibitory myelin and neuronal debris, it also upregulates the local production of neurotrophins [36]. Consistent with this, nerve regeneration after crush injury of the sciatic nerve was found to be worse in Tlr2^−/−^, Tlr3^−/−^, Tlr4^−/−^, Tlr5^−/−^, and Tlr7^−/−^ mice than in C57BL/6 mice, as demonstrated by histomorphometric measurements and CatWalk gait analyses. Notably, Tlr-knockout mice exhibit a much more prominent fiber debris area and significantly down-regulated expression of a subset of myelination-related factors. Thus, the expression of myelination-related transcription factors and neurotrophin genes is reduced, and nerve regeneration is impaired in Tlr-knockout mice following sciatic nerve crush injury.

There are 10 types of TLRs in humans, but 13 types in rats. In a study using male Sprague–Dawley rats, recovery of the facial nerve and expression of TLR mRNA and protein, determined using RT-PCR and Western blotting, respectively, were assessed after compression or transection of the facial nerve [41,42]. The scores for whisker movements of the vibrissae muscle and blink reflexes of the eyelid were significantly lower in both cutting and crush groups (*p* < 0.05) than in the control group on day 4 after injury. There were no significant differences between the crush group and the control group 14 days and 3 months after the injury, indicating restoration of facial nerve function. However, in rats subjected to a cutting injury, the score remained significantly lower in the cutting group compared with the control group at 14 days and 3 months, and there was no recovery of facial nerve function. RT-PCR analyses of TLR1–13 mRNA expression patterns on day 4 after injury showed higher expression of TLR2, -6, and -7 in compression and cutting groups than in the corresponding control groups. On day14 after injury, TLR2, -6, and 7- mRNA expression trended higher in the compression group than in the control group, a difference that did not reach statistical significance. TLR1–7, -9, and -10 mRNA expression also trended higher at this time point in the cutting group than in the control group, although apart from TRL2 mRNA, these differences were not statistically significant. On day 4 after injury, TLR9 and -13 mRNA levels were significantly lower in both compression and cutting groups compared with the corresponding control groups (*p* < 0.05). In nerve transection experiments, only the expression of TLR2 mRNA remained significantly higher in the cutting group than in the control group on day 14 after injury (*p* < 0.05). Thus, although all TLR mRNAs were expressed after facial nerve injury, changes in the expression of different TLRs varied depending on the facial nerve injury method and the post-injury period [43]. At the protein level, Western blot analyses showed increases in TLR11 and -13 in the nerve-crush group on day 14 after injury and increases in TLR1, -2, -3, -4, -5, -8, -10, -11, -12, and -13 in the cutting group (*p* < 0.05). Three months after facial nerve injury, TLR10 and -11 increased in the nerve-crush group, and TLR1, -4, -5, -8, -11, and -12 increased in the cutting group (*p* < 0.05). Collectively, these results confirm changes in TLR1–13 mRNA expression in the peripheral nerve area after facial nerve injury, implicating the expression of TLRs in the process of facial nerve injury and regeneration and suggesting the specific association of TLR10, -11, and -13 with recovery from facial palsy [44]. Few studies have focused on the roles of TLR10–13 in the nervous system. This is particularly the case for TLR11–13 because these receptors are not found in humans. However, studies in mice have demonstrated that TLR11–13 are expressed in astrocytes, microglia, and neuronal cells of the central nervous system and are increased during parasite infection. Activation of neurons, astrocytes, and endothelial cells of blood vessels through TLR11–13 has been reported to improve neuroprotective function, suggesting the need for further research in this area. 

## 4. Conclusions

Studies have indicated that TLR1–13 is involved in the degeneration and regeneration of the facial nerve following facial nerve compression or transection. TLR10, TLR11, and TLR13 increased during compression injury of the facial nerve, whereas during cutting injury, TLR1–5, TLR8, TLR10, and TLR11–13 increased, indicating that these TLRs were involved in degeneration and regeneration of the facial nerve during each type of injury. Expression of TLRs during facial nerve damage was found to contribute to nerve regeneration as well as to the immune response to inflammation and by-products of the damaged nerve. Extensive research is currently being conducted to understand the factors that affect recovery and nerve regeneration after nerve injury, but innovative treatment technologies have not yet been developed. In-depth research on the roles of TLRs in the damage and regeneration of the facial nerve may provide basic data on facial nerve recovery that can be used to develop improved treatments for facial nerve palsy, as well as for damaged peripheral and central nerves.

## Data Availability

Not applicable.

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
