# Peer review of "Expression and Role of Toll-like Receptors in Facial Nerve Regeneration after Facial Nerve Injury"

_ijms, 2023, doi:10.3390/ijms241411245_

Round 1

Reviewer 1 Report

The manuscript " Expression and role of Toll-like receptors in facial nerve regeneration after facial nerve injury" is a review that aims to deepen and relate the literature concerning the role of Toll-like receptors (TLRs) in regeneration following facial nerve injury .

The work is not well structured, starting from the title where only the topic of facial nerve regeneration appears, which is then mixed in the core text with a more generic nerve regeneration (sciatic nerve injury) and even spinal cord injury.

The review does not sufficiently elaborate on the topic of facial nerve regeneration, for this reason the authors should consider these major revisions:

- Changing the title of the review, highlighting that the study is focused on nerve/spinal injury and regeneration, in this way it will be possible to update the literature, which is very old.

- The Figure 1 is not very significant, only the drawings of the receptors appear, but no tools showing upregulation or downregulation following injury.

If they want to keep the image, the authors must insert clearer schemes that make visible what is stated in paragraph 3.

- Changing the introduction, which can no longer start with the facial palsy, but maybe with the topic of nerve regeneration and the differences between the central and peripheral nervous system, given that authors want to keep spinal cord injury in the review.

- The conclusions are poor, they do not draw any message, no final consideration.

Author Response

We would like to thank the editor and reviewers for their helpful comments and suggestions, which we have incorporated into our revised manuscript. Our point-by-point responses to specific reviewer comments are provided below.

Reviewer 2 Report

This review by Lee JM et al. provides a comprehensive analysis of the role of Toll-like receptor (TLR) signaling in nerve regeneration, with a focus on animal studies that investigate the contribution of different TLRs and their signaling pathways to the regenerative process. While the review is overall interesting and summarizes relevant studies, there are areas where clarity and focus could be improved. Specifically, the abstract, introduction, and general TLR paragraph lack focus and would benefit from careful revision. 

Additionally, the following major comments should be addressed:

1.    The introduction section should include an introductory paragraph about the etiology of facial palsycovering the potential causes, the disease development, the pathogenesis, and the current treatment options.

2.    It is necessary to include a section discussing TLR expression and roles in CNS resident cells, particularly neurons. This addition will help readers better understand the role of TLRs in the context of nerve injury.

3.    The abstract needs a complete revision, including an introduction to the general topic of facial nerve injury, potential causes, and current treatments. The current structure of the abstract (objective, methods, results, and conclusion) should be removed as it is unnecessary and misleading. A more detailed abstract will help readers better understand the scope of the review and the specific topics that will be discussed.

4.    Citations should be placed at the end of each statement rather than only at the end of paragraphs. This will make it clearer where the author found the information and improve the organization of the literature.

5.    In line 69, the author states that human patient data is reviewed and discussed in the work. However, it is unclear if any human studies are discussed throughout the manuscript, as most of the studies were carried out in animal models. The authors should consider discussing the animal work in the context of the human system or remove the claim that human studies have been reviewed.

6.    Figure 1 and Table 1 are not mentioned in the text, except for the subsection title in line 128. Both the table and figure should be mentioned and carefully discussed in the main text ,especially as they represent relevant injury models that are further discussed throughout the rest of the manuscript. 

In addition to the major comments, the following minor comments should be addressed:

1.    The title mentions facial nerve, but the review also includes studies on sciatic nerve and spinal cord injury. The title should be revised to encompass nerve injury and Toll-like receptor signaling more broadly.

2.    In line 52, the statement "DAMPs which promote expression of TLR" might be referring to the activation of the TLR-signaling pathway instead of TLR expression.

3.    Line 54 is unclear whether the authors are referring to the activation of the TNF pathway or the production of TNF protein.

4.    In line 58-60, the authors discuss "day post-injury," but it is unclear which system is being referred to, such as in vitro, ex vivo, human, or mouse. This information is necessary for understanding the context. 

The abstract, introduction, and general TLR introduction paragraph would benefit from a careful revision on content and english as some passages are hard to understand. 

Author Response

(The authors gave the same response as above.)

Round 2

Reviewer 1 Report

The manuscript, after an extensive review by the authors and having focused more on the role of TLRs on the peripheral nervous system, of which the facial nerve is a part, is now, in my opinion, worthy of publication in the present form.

Reviewer 2 Report

I am pleased with the author's thorough addressing of the questions and concerns I raised in my initial review. The revised manuscript has undergone significant improvements to enhance readability and ensure a smooth flow throughout.